# Prostate Cancer Semantic Segmentation by Gleason Score Group in bi-parametric MRI with Self Attention Model on the Peripheral Zone

**Audrey Duran**[1]                                    AUDREY.DURAN@CREATIS.INSA-LYON.FR
[1] *Univ Lyon, INSA-Lyon, Université Claude Bernard Lyon 1, UJM-Saint Etienne, CNRS, Inserm,*
*CREATIS UMR5220, U1206, F69621 LYON, France*

**Pierre-Marc Jodoin**[2]                          PIERRE-MARC.JODOIN@USHERBROOKE.CA
[2] *Computer Science Department, University of Sherbrooke, Sherbrooke, QC, Canada*

**Carole Lartizien**[1]                            CAROLE.LARTIZIEN@CREATIS.INSA-LYON.FR

**Editors:** Under Review for MIDL 2020

## Abstract

In this work, we propose a novel end-to-end multi-class attention network to jointly perform peripheral zone (PZ) segmentation and PZ lesions detection with Gleason score (GS) group grading. After encoding the information on a latent space, the network is separated in two branches: 1) the first branch performs PZ segmentation 2) the second branch uses this zonal prior as an attention gate for the detection and grading of PZ lesions. The model was trained and validated with a 5-fold cross-validation on an heterogeneous series of 98 MRI exams acquired on two different scanners prior prostatectomy. In the free-response receiver operating characteristics (FROC) analysis for clinically significant lesions (defined as GS > 6) detection, our model achieves 75.8% ±3.4% sensitivity at 2.5 false positive per patient. Regarding the automatic GS group grading, Cohen's quadratic weighted kappa coefficient is 0.35 ±0.05, which is considered as a fair agreement and an improvement with regards to the baseline U-Net model. Our method achieves good performance without requiring any prior manual region delineation in clinical practice. We show that the addition of the attention mechanism improves the CAD performance in comparison to the baseline model.

**Keywords:** semantic segmentation, attention model, convolutional neural network, computer-aided detection, magnetic resonance imaging, prostate cancer

## 1. Introduction

Multiparametric magnetic resonance imaging (mp-MRI), which combines T2-weighted imaging with diffusion-weighted, dynamic contrast material–enhanced, and/or MR spectroscopic imaging, has shown promising results in the detection of prostate cancer (PCa). However, characterizing focal prostate lesions in mp-MRI sequences is time demanding and challenging, even for experienced readers, especially when individual MR sequences yield conflicting findings. There has been a considerable effort, in the past decade, to develop computer aided detection and diagnosis systems (CAD) of PCa cancer as well as prostate segmentation (Wildeboer et al., 2020).

The vast majority of developed CAD models focus on the detection and segmentation of clinically significant (CS) cancers, defined here as lesions with Gleason score > 6 (Ploussard

et al., 2011), where the Gleason score (GS) characterizes the cancer aggressiveness. The current lesion grading system is based on GS groups, where GS 7 is separated in two groups (GS 3+4 and GS 4+3) and GS 9 and GS 10 are gathered in the same group (Epstein et al., 2016). Despite an important improvement brought by CAD systems that automatically map CS PCa lesions, there is a need to go one step further by also predicting the degree of PCa aggressiveness. The binary prediction does indeed not suffice for active surveillance of patients with low aggressiveness cancers or patients that could benefit from focal therapy. This topic has been recently addressed by a few studies, using deep learning methods (Cao et al., 2019; de Vente et al., 2019; Abraham and Nair, 2018; Tsehay et al., 2017).

Moreover, most of the current CAD systems rely on a 2-steps workflow, with a preliminary step to focus on the prostate. Recent studies demonstrate good performance of architectures combining two cascaded networks: one segmenting or performing a crop around the prostate, followed by a binary PCa lesion segmentation model (Yang et al., 2017; Wang et al., 2018; Hosseinzadeh et al., 2019).

In this work, we propose a novel end-to-end deep architecture called ProstAttention-Net that automatically performs segmentation of the prostate peripheral zone and uses this zonal prior as an attention gate for the detection and grading of peripheral zone (PZ) lesions. We show that the addition of this attention mechanism improves the CAD performance.

The main contributions can be summarized as follows:

- A novel end-to-end architecture that performs jointly multi-class segmentation of PCa lesions and PZ segmentation

- Evaluation based on the FROC metric at the GS group level

- Evaluation on a heterogeneous PCa database of 98 patients (1.5T and 3T scanners from 2 manufacturers) with whole-mount histopathology slices of the prostatectomy specimens as ground truth

## 2. Materials and methods

### 2.1. Data description

The dataset consists in a series of bi-parametric MRI (bp-MRI) images from 98 patients, acquired in clinical practice at our partner clinical center. All patients were scheduled to undergo radical prostatectomy and gave written informed consent for the use for research purposes of their MR imaging and pathologic data.

Imaging was performed on two different scanners from different constructors: 57 exams were acquired on a 1.5 Tesla scanner (Symphony; Siemens, Erlangen, Germany) and 41 on a 3 Tesla scanner (Discovery; General Electric, Milwaukee, USA). In this study, we included axial T2 weighted (T2w) and apparent diffusion coefficient (ADC) maps computed by the constructor from the diffusion weighted imaging (DWI) sequence. Parameters of the sequences are reported in Table 1.

MRI exams were reviewed in consensus by two uroradiologists who outlined PZ focal lesions that showed abnormal signal on images. The prostatectomy specimens were analyzed *a posteriori* by an anatomopathologist thus providing the histological ground truth as seen on Figure 1. After correlation with the whole-mount specimens, the uroradiologists reported

Table 1: Parameters for prostate imaging on the 1.5T and 3T scanners

| Scanner field | Sequence | $T_R$ (ms) | $T_E$ (ms) | FOV (mm) | Matrix (voxels) | Voxel dimension (mm) |
|---|---|---|---|---|---|---|
| 1.5T | T2w | 7750 | 109 | $200 \times 200$ | $256 \times 256$ | $.78 \times .78 \times 3$ |
| 3T | T2w | 5000 | 104 | $220 \times 220$ | $512 \times 512$ | $.43 \times .43 \times 3$ |
| 1.5T | ADC | 4800 | 90 | $300 \times 206$ | $128 \times 88$ | $2.34 \times 2.34 \times 3$ |
| 3T | ADC | 5000 | 90 | $380 \times 380$ | $256 \times 256$ | $1.48 \times 1.48 \times 3$ |

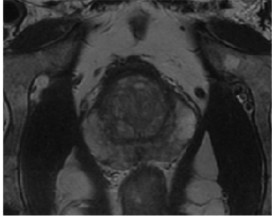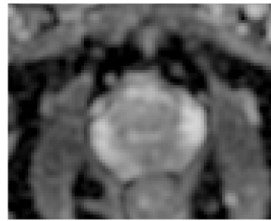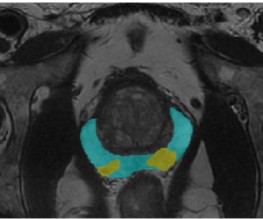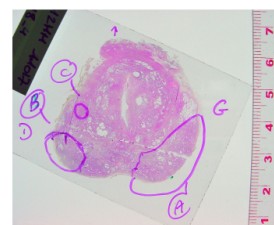

Figure 1: Prostate MRI annotation. From left to right: T2w MR image, ADC maps, final PZ annotation overlapped on T2w and histology slice.

132 lesions of the peripheral zone. Their distribution according to the GS group is detailed in Table 2. Lesions that were not visible on mp-MRI exams *a posteriori* were not reported. In addition to lesions annotation, the radiologists also contoured the peripheral zone on each volume.

Table 2: Lesions distribution by Gleason Score

| GS 3+3 | GS 3+4 | GS 4+3 | GS 8 | GS $\geq$ 9 | Total |
|---|---|---|---|---|---|
| 37 | 47 | 23 | 16 | 9 | 132 |

## 2.2. ProstAttention-Net for PCa and PZ segmentation

As depicted on Figure 2, our ProstAttention-Net model is an end-to-end multi-class deep network that jointly performs two tasks: 1) the segmentation of the PZ and 2) the detection and GS group grading of PZ lesions.

The encoder of our network first encodes the information from multichannel T2w and ADC input images into a latent space. This latent representation is then connected to two decoding branches: 1) the first one performs a binary PZ segmentation 2) the second one uses this zonal prior as an attention mechanism for the detection and grading of PZ lesions. This is somewhat inspired but different from the Attention U-Net (Schlemper et al., 2019), as our attention map comes from the resampling of the PZ prediction instead of the

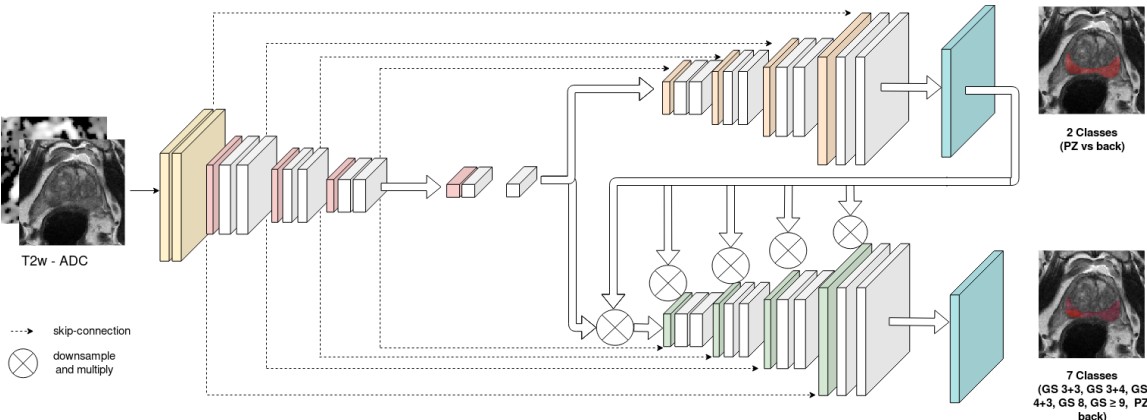

Figure 2: Our proposed ProstAttention-Net model. On the left is the encoder, on the top right is the PZ decoder and on the bottom right is the lesion decoder. The encircled X stands for the attention operation.

preceding convolutional block. The idea here is to enforce the tumor prediction to be within the PZ by shutting down the neurons located outside the PZ.

The output probabilistic map of the PZ decoder serves as a soft attention map to the lesion decoder branch. This self trained attention map is first downsampled to the resolution of each block of the lesion branch and multiplied to the input feature maps of this block. Our attention operation is a channel-wise Hadamard product between the resampled output of the PZ branch and the feature maps of the second branch. The backbone CNN architecture of our attention model is based on U-Net (Ronneberger et al., 2015), with batch normalization (Ioffe and Szegedy, 2015) layers to reduce over-fitting. The encoder part of our model contains five blocks, each composed of two convolutional layers with kernel size $3 \times 3$, followed by a Leaky ReLu (Maas et al., 2013) and MaxPool layers. The PZ decoder branch follows the same architecture but with transposed convolutions to increase the feature maps resolution. Its output has two channels corresponding to the PZ and background classes. The lesion decoder has an architecture similar to the PZ decoder branch except that it produces 7-channels segmentation maps, corresponding to the five Gleason Score groups, PZ and background classes.

The global loss of the ProstAttention-Net is defined as

$$L = \lambda_1 . L_{PZ} + \lambda_2 . L_{lesion} \tag{1}$$

where $L_{PZ}$ and $L_{lesion}$ are the losses corresponding to the PZ and lesion segmentation task and $\lambda_1$ and $\lambda_2$ are weights to balance between both losses whose value can be varied during training. The $L_{PZ}$ and $L_{lesion}$ loss functions were defined as the sum of the cross entropy and dice losses. To deal with class imbalance, each loss term was weighted by a class-specific weight $w_c$. These losses can be detailed as:

$$L_{PZ} = 1 - 2\frac{\sum_{c=1}^{2} w_c \sum_{i=1}^{N} y_{ci}p_{ci}}{\sum_{c=1}^{2} w_c \sum_{i=1}^{N} y_{ci} + p_{ci}} - \frac{1}{N}\sum_{i=1}^{N}\sum_{c=1}^{2} \mathbb{1}_{y_i \in C_c} w_c \log p_{ci} \qquad (2)$$

$$L_{lesion} = 1 - 2\frac{\sum_{c=1}^{7} w_c \sum_{i=1}^{N} y_{ci}p_{ci}}{\sum_{c=1}^{7} w_c \sum_{i=1}^{N} y_{ci} + p_{ci}} - \frac{1}{N}\sum_{i=1}^{N}\sum_{c=1}^{7} \mathbb{1}_{y_i \in C_c} w_c \log p_{ci} \qquad (3)$$

where $w_c$ is the class-specific weight, $p_{ci}$ the probability predicted by the model for the observation $i$ to belong to class $c$ and $y_{ci}$ is the ground truth label for pixel $i$.

### 2.3. Experiments

To assess the impact of the attention module, our model is compared to a U-Net baseline method. This U-Net consists of the backbone architecture of our model, with batch normalization and the same number of convolutional layers. The output segmentation map consists of a 7-channels output, similar to the lesion branch output. To enable a fair comparison, data pre-processing, validation and hyperparameter search was performed the same way as for our attention model.

#### 2.3.1. IMPLEMENTATION DETAILS

The whole network was trained end-to-end using Adam (Kingma and Ba, 2014) with an initial learning rate of $10^{-3}$ and a L2 weight regularization with $\lambda = 0.001$. Input T2w and ADC images were resampled to a $1 \times 1 \times 3$ mm voxel size and automatically cropped to a $96 \times 96$ pixels region on the image's center. Intensity was standardized to scale pixel values to have a zero mean and unit variance. Data augmentation was applied during the training phase to reduce overfitting.

Concerning loss weights defined in Equation (1), $\lambda_1$ was set to 1 and $\lambda_2$ to 0 during the first 15 epochs in order to focus the training on the PZ segmentation task. Then, $\lambda_2$ was changed to 1 to allow an equal contribution of the $L_{PZ}$ and $L_{lesion}$ loss terms. Class-specific weights $w_c$ were set to 0.002 for background, 0.14 for PZ and 0.1715 for each lesion class.

The pipeline was implemented in python with the Keras-Tensorflow library (Chollet et al., 2015; Abadi et al., 2015). Training and testing were performed on a NVIDIA GeForce GTX 1660 Ti with 6GB memory Graphics processing unit.

#### 2.3.2. PERFORMANCE EVALUATION

Training and evaluation were conducted on the whole patient 3D volumes, each constituted of 24 transverse slices including slices without visible prostate or without lesions annotation. No PZ mask was applied to the input bp-MRI images or output prediction maps, both during training or evaluation phases. Both models were trained and validated using a 5-fold cross-validation. Each fold contained 19 or 20 patients (i.e. around 460 slices) and was balanced regarding its lesion classes.

Lesion detection performance was evaluated through free-response receiver operating characteristics (FROC) analysis. FROC curves report detection sensitivity at the lesion level as a function of the mean number of false positive lesion detections per patient. Lesion maps were estimated from the labeled maps outputted from the lesion decoder branch using

a 3-connectivity rule to identify the connected components. Lesions smaller than 9 voxels (ie. 27 mm$^3$, the size of the smallest annotation) were removed. Each detected lesion was assigned a *lesion probability score* corresponding to the average of the voxel probability values in the cluster. FROC curve was then plotted by varying a threshold on this lesion probability score. For each value of this threshold, a lesion was considered as a true positive if at least 10% of its volume intersected a true lesion and if its lesion probability score exceeded the threshold. If it did not intersect a true lesion, it was considered as a false positive.

Two different FROC analysis were performed, the first one evaluating the performance of the models to discriminate clinically significant lesions (GS > 6), the second one evaluating its ability to discriminate lesions of each class of each GS group.

To evaluate the agreement between the ground truth and the 7-classes prediction, we used the Cohen's quadratic weighted kappa coefficient of agreement, as proposed for GS grading in PROSTATEx-2 challenge. Cohen's kappas takes into account the class-distance between the ground truth and the prediction. In this study, it was computed at voxel-level considering the 5 GS groups and the PZ, thus ignoring the background class.

## 3. Results

Figure 3 shows predictions maps from the U-Net baseline model and our ProstAttention-Net. In the first case, U-Net could not detect the GS 4+3 lesion, even if it was able to discriminate this region from normal PZ tissue (this region was labeled as background). ProstAttention-Net, thanks to the prior prostate zonal prediction, knew this region was belonging to the PZ and could not only classify this region as a lesion but also assign it the correct GS. In the second case, U-Net identified the GS 3+4 lesion but not in the correct class and with a broad delineation. ProstAttention-Net could delineate the region in a finer way and assign it the correct GS. On the third and on the last example, U-Net predicted a region outside of PZ. This is the kind of cases that we are avoiding with ProstAttention-Net, which focuses attention on the PZ. On the last example, ProstAttention-Net improves lesion segmentation in comparison to U-Net, as the lesion is mostly assigned to the correct class GS 3+4. Also note that part of the lesion is mistakenly assigned to GS 8, which is an interesting result since GS 8 is composed of grade 4 cells only while GS 3+4 has a majority of grade 3 cells but also some grade 4 cells. PZ prediction is also not perfect on this difficult case.

Figure 4 shows the ProstAttention-Net FROC results for CS (GS > 6) cancer detection in comparison to the U-Net model. Our model achieves 75.8% ±3.4% sensitivity at 2.5 false positives per patient, while U-Net baseline model reaches 66% ±18.5% sensitivity at 2.5 false positives per patient.

Our attention model enables to reduce the average number of false positives per patient by limiting the area where a lesion should be searched. The green curve shows ProstAttention-Net performance if, instead of considering the whole 24-slices volume, we only examine slices with at least one annotated lesion, as in Cao et al.'s work (Cao et al., 2019). This curve shows that the performance of ProstAttention-Net is similar to FocalNet model (Cao et al., 2019) trained with a cross entropy loss on a dataset 4 times larger than our's and acquired on 3T scanners from one manufacturer only. For a 1 false positive per

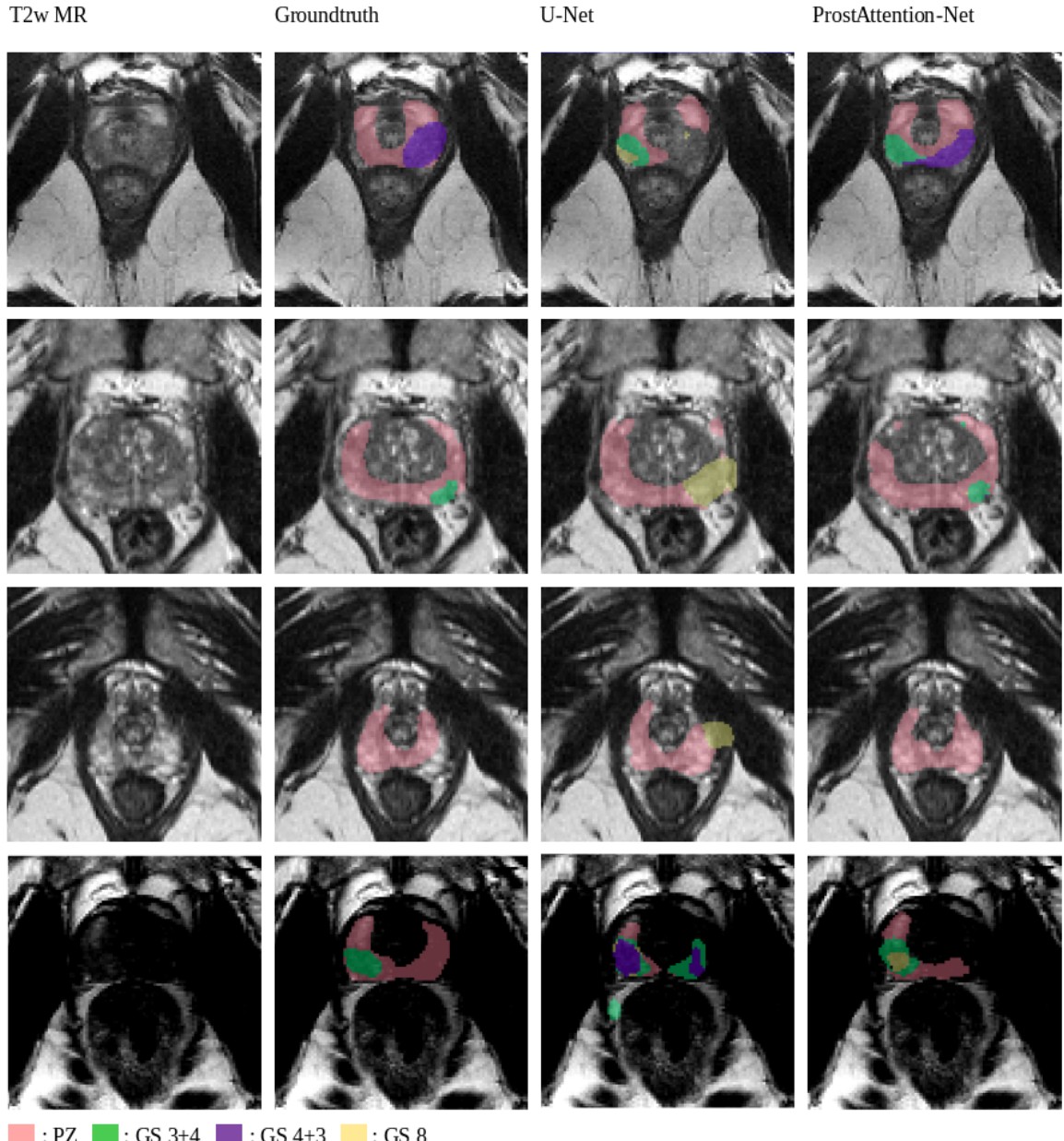

Figure 3: Prediction comparison for several images from the validation set. The images from the first and last row comes from a 3T scanner whereas the other ones come from a 1.5T scanner.

patient, FocalNet with a cross entropy loss reached $\sim 60\%$ just like ProstAttention-Net (FocalNet reaches $\sim 80\%$ with a focal loss).

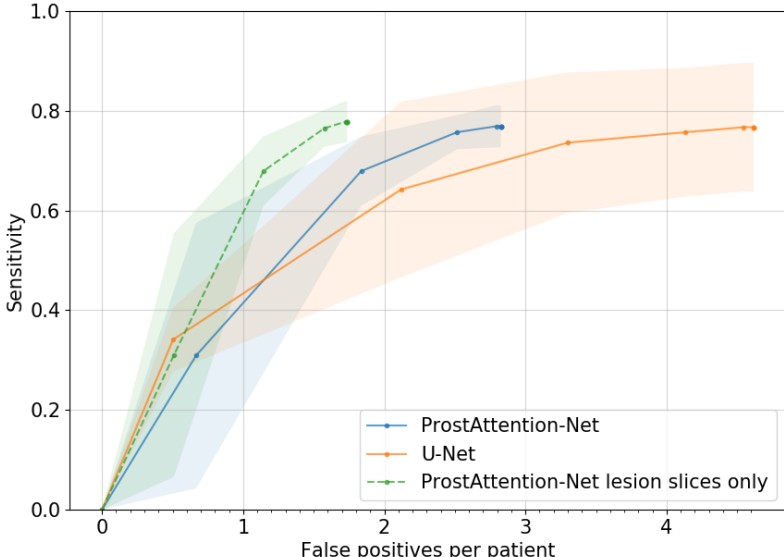

Figure 4: FROC analysis for detection sensitivity on clinically significant lesions (GS > 6), based on 5-fold cross-validation. Solid lines show performance on the whole 24-slices volume while the dotted line shows results considering only slices with at least one lesion. The transparent areas are 95% confidence intervals corresponding to 2x the standard deviation.

Results of the FROC analysis for each GS group are presented in Table 3. Due to the limited number of lesions per class and fold, these results are to be considered with care. They show that detection performance for each GS group is globally correlated to the lesion aggressiveness: the higher the cancer GS group, the better the detection performance, as also reported in Cao et al. (2019). However, we can observe that ProstAttention-Net performance for the GS 8 are under GS 4+3 one's. The confusion matrix (data not shown) that serves as the basis to compute the kappa coefficient indicates that GS 8 are often confused with GS 4+3 lesions. We can hypothesize that our model does not perfectly disentangle these two overlapping classes, most likely because of a lack of GS 8 training samples. Comparing to U-Net model, except for GS 8, our method outperforms the baseline.

Cohen's quadratic weighted kappa coefficient also reflects the capacity of our model to segment correctly the different GS groups and PZ. We obtain a coefficient of 0.35 ±0.05 with ProstAttention-Net, while the corresponding kappa coefficient for U-Net is 0.31 ±0.08. This reflects the contribution of the attention gate in segmenting the different classes. In order to compare our kappa score to study by de Vente et al. (2019) where multi-class segmentation of PCa cancer is performed with a U-Net, we computed another kappa score. Assuming that the voxel-wise quadratic-weighted kappa score (reported in Table 1 of de Vente et al. (2019)) is computed considering 5 GS groups (including GS 3+4, GS 4+3, GS 8, GS $\geq$

Table 3: Comparison between our ProstAttention-Net and U-Net detection sensitivity at given false positive (FP) per patient thresholds on each Gleason Score group.

| | GS $\geq$ 9 | | GS 8 | | GS 4+3 | | GS 3+4 | | GS 3+3 | |
|---|---|---|---|---|---|---|---|---|---|---|
| | 1FP | 1.5FP | 1FP | 1.5FP | 1FP | 1.5FP | 1FP | 1.5FP | 1FP | 1.5FP |
| U-Net | 0.70 | 0.70 | **0.43** | **0.45** | 0.40 | 0.50 | 0.43 | 0.47 | 0.17 | 0.17 |
| ProstAttention-Net | **0.80** | **0.80** | 0.28 | 0.28 | **0.48** | **0.54** | **0.46** | **0.54** | **0.19** | **0.25** |

9, and a normal group gathering background, healthy prostate and GS 6), the equivalent kappa for our study gives 0.463 ±0.033 for U-Net and 0.538 ±0.032 for ProstAttention-Net. This compares favorably to the kappa score of 0.391 ±0.062 achieved by their best model (SLOR).

## 4. Discussion

Our ProstAttention-Net model outperforms a well-tuned U-Net at the task of detecting clinically significant (GS $>$ 6) PZ cancers. It succeeds in learning from an heterogeneous database containing T2 and ADC exams acquired on scanners from two different manufacturers, with different static magnetic fields (1.5T and 3T), sequence and reconstruction parameters. Performance that we report in this study is a good indicator that our model is robust to the heterogeneity of the training database. This is the first time a deep convolutional network trained on an heterogeneous dataset can successfully localize prostate lesions as well as predict their Gleason score. This is encouraging for the generalization capacity of our model, which is one important CAD limitation nowadays. We provided elements of comparison with state-of-the art methods, however a fair comparison with some previous works is difficult as the datasets and evaluation metrics are not the same. For a better comparison, it would be interesting to evaluate our model on the PROSTATEx-2 dataset which we intend to do in the near future.

In order to enhance the performance, we plan to include additional MRI exams in the training database. A semi-supervised approach will also be considered to allow the inclusion of clinical data with partial annotations. The contribution of additional modalities (high b-value diffusion MR, perfusion) should also be studied. Furthermore, it could be valuable to use a sensitivity / specificity loss or ranking based loss, as in Cao et al. (2019), to fully exploit the existing hierarchy between each GS group. Finally, an approach by Gleason pattern as in Azizi et al. (2017) might be studied.

## Acknowledgments

This work was supported by the RHU PERFUSE (ANR-17-RHUS-0006) of Université Claude Bernard Lyon 1 (UCBL), within the program "Investissements d'Avenir" operated by the French National Research Agency (ANR). We thank Pr. Olivier Rouvière

and Tristan Jaouen for their expertise and contribution in building the 3D MRI prostate database.

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
