# OpenReview forum: "Prostate Cancer Semantic Segmentation by Gleason Score Group in bi-parametric MRI with Self Attention Model on the Peripheral Zone"
_MIDL.io/2020/Conference — MIDL 2020_

### Official Review · AnonReviewer4 · 2020-03-13
**Valid clinical problem, however, the framing of problem is impaired.**

**Rating:** 2
**Confidence:** 5
**Recommendation:** Poster

**Summary:**

The paper present a method for prostate cancer grading (aka. semantic segmentation of prostate cancer to different Gleason score group) in MRI images. The paper is focused on grading in peripheral zone which is considered as a valid clinical interest. The proposed solution also segment the boundaries of peripheral zone region and use this information as an attention mechanism to perform the following detection and grading using high level latent features of the second part of the deep network. The also validate the methods by performing 5 folds cross validation.

**Strengths:**

** Quality of evaluations:
- Author provides results in 5 folds cross validation settings and also they have provided adequate visual results.
** Clarity and Relevance:
- The problem is clinically relevant and clinical dataset well described.
- The paper is well-written, and the experimental setup is convincing, and in general, it is easy to follow the paper.


**Weaknesses:**

** Justification needs for the choice of label interpretation:
- From the pathological view point, the pathology report of GS x+y representing a tissue area with the combined pattern of Gleason x and y where Gleason x is prominent in the regions (e.g. Gleason 3+4 means we are dealing with a region with larger area of Gleason 3, and some smaller areas of Gleason 4.). Combining our knowledge from the biology and machine learning, one key take away here is that from the machine learning perspective the multi-class problem that we are dealing with here, is a multi-instance learning problem. “Instead of receiving a set of instances which are individually labeled, the learner receives a set of labeled bags, each containing many instances. [Wikipedia]”. This is the beauty of machine learning for medical imaging though, to combine knowledge from biology, physics of modalities, and also machine learning.
- So, based on the above point, and based on the very heterogeneous nature of prostate tissue, learning the difference between class GS 3+3, 3+4, and 4+3 and 4+4 can be address as a simple multi-class problem “properly not blindly” either by using some especial arrangements as it has been previously proposed in [1] (learning deep features manifold, clustering), or having a very large dataset including enough rare samples (i.e. proper distribution of higher grades and lower grade) to perform a multi-instance learning or multi-class classification.
- To back up my arguments, I just refer authors to their own results presented in Figure 3, last row. In the GT we have a case of GS 3+4 (i.e. large region of Gleason pattern 3, some smaller region of Gleason pattern 4). Authors predict large area of GS 3+4 and smaller region of GS 4+4/GS 8, which is absolutely interesting. Let’s break this down. Authors predict smaller region of only Gleason pattern 4, and larger region of Gleason pattern 3 (i.e. GS 3+4). Simply what I am seeing here, is that, your features is actually working fine, the feature manifold that you have learned as GS 3+4, is actually the manifold of Gleason pattern 3.
- I am fairly confident by performing some unsupervised method on top of your learn features, you will get the performance boost.

** Missing key references:
 [1] Azizi et al. "Detection and grading of prostate cancer using temporal enhanced ultrasound: combining deep neural networks and tissue mimicking simulations." International journal of computer assisted radiology and surgery 12.8 (2017): 1293-1305.
[2] Azizi, et al. "Classifying cancer grades using temporal ultrasound for transrectal prostate biopsy." International Conference on Medical Image Computing and Computer-Assisted Intervention. Springer, Cham, 2016.


**Detailed Comments:**

- Please provide a reference for considering GS greater than 6 (3+3) as a clinically significant cancer. In some literature clinically significant cancer is defined at the border of GS 3+4 vs GS 4+3.
- Please see above.

*** Some other suggestions to improve the work for future:
- It would be great to consider providing the location and distribution of samples in the data and analyze its effect in the prediction results, especially if you also can includes some samples from central zone, so you can contrast them properly. ( i.e. peripheral or central zone, and also base, apex, and mid-gland )
- I highly recommend re-framing of the problem as suggested and analyzed by Azizi et al., or considering multi-instance learning solutions if you think you can have access to enough samples.


**Justification Of Rating:**

Overall, this paper describes an extension of an existing approach and of an incremental nature. Also, authors have overlooked the definition of Gleason scores meaning from the pathological perspective which I believe results in an impaired but yet working solution.

**Paper Type:**

both

**Special Issue:**

no

---

> ### Author Response · Authors · 2020-03-27
> **Reply to the reviewer - Rebuttal**
>
> 1)  Justification needs for the choice of label interpretation and missing key references by Azizi et al.
>
> Thanks for sharing this interesting paper by Azizi et al.
> We indeed assume that each lesion contains an homogeneous volumetric distribution of both Gleason x and Gleason y (following the reviewer notations), so that this lesion can be attributed a unique GS x+y  score.  Work by Cao et al. (among others) demonstrated good performance from mp-MRI data under the assumption of homogenous GS per lesion.  In the near future, we will increase the size of our dataset to confirm the trend that we observed and to enable a quantitative comparison with the work by Cao et al. so as to establish a performance baseline. We will then build on more advanced strategies, following the above mentioned ideas. The suggestion to perform clustering based on the deep feature, following the idea proposed by Azizi et al. is interesting. This point will be shortly (due to size limitation) addressed in the revised version of the paper.
>
> 2) Provide a reference for considering GS greater than 6 (3+3) as a clinically significant cancer.
>
> The notion of clinically significant cancer varies among papers. Our definition comes from the paper  “The Contemporary Concept of Significant Versus Insignificant Prostate Cancer” by Ploussard et al.  Furthermore, we wanted to be able to compare to state-of-the-art papers - like Cao et al., 2019, Wang et al., 2019, Abraham et al., 2018 - in which clinically significant lesions are those with GS>6. Also, in PROSTATEx challenge, clinically significant lesions were defined as lesions with a GS of 7 or higher. These references will be added in the revised version of the paper.

---

### Official Review · AnonReviewer1 · 2020-03-15
**Prostate Cancer Semantic Segmentation by Gleason Score Group in mp-MRI with Self Attention Model on the Peripheral Zone**

**Rating:** 4
**Confidence:** 5

**Summary:**

Well described clinical problem definition of prostate MRI.
Self-attention is an interesting strategy. Grouping it in different ISUP groupings is challenging.

Prostate Cancer Semantic Segmentation by Gleason Score Group in mp-MRI with Self Attention Model on the Peripheral Zone
Prostate Cancer Semantic Segmentation by Gleason Score Group in mp-MRI with Self Attention Model on the Peripheral Zone
Prostate Cancer Semantic Segmentation by Gleason Score Group in mp-MRI with Self Attention Model on the Peripheral Zone
Prostate Cancer Semantic Segmentation by Gleason Score Group in mp-MRI with Self Attention Model on the Peripheral Zone
Prostate Cancer Semantic Segmentation by Gleason Score Group in mp-MRI with Self Attention Model on the Peripheral Zone


**Strengths:**

Well described clinical problem definition of prostate MRI.
Self-attention is an interesting strategy. Grouping it in different ISUP groupings is challenging.


Well described clinical problem definition of prostate MRI.
Self-attention is an interesting strategy. Grouping it in different ISUP groupings is challenging.
Well described clinical problem definition of prostate MRI.
Self-attention is an interesting strategy. Grouping it in different ISUP groupings is challenging.
Well described clinical problem definition of prostate MRI.
Self-attention is an interesting strategy. Grouping it in different ISUP groupings is challenging.
Well described clinical problem definition of prostate MRI.
Self-attention is an interesting strategy. Grouping it in different ISUP groupings is challenging.


**Weaknesses:**

Not a lot of data.

Well described clinical problem definition of prostate MRI.
Self-attention is an interesting strategy. Grouping it in different ISUP groupings is challenging.
Well described clinical problem definition of prostate MRI.
Self-attention is an interesting strategy. Grouping it in different ISUP groupings is challenging.
Well described clinical problem definition of prostate MRI.
Self-attention is an interesting strategy. Grouping it in different ISUP groupings is challenging.
Well described clinical problem definition of prostate MRI.
Self-attention is an interesting strategy. Grouping it in different ISUP groupings is challenging.


**Detailed Comments:**

Well described clinical problem definition of prostate MRI.
Self-attention is an interesting strategy. Grouping it in different ISUP groupings is challenging.
Well described clinical problem definition of prostate MRI.
Self-attention is an interesting strategy. Grouping it in different ISUP groupings is challenging.
Well described clinical problem definition of prostate MRI.
Self-attention is an interesting strategy. Grouping it in different ISUP groupings is challenging.
Well described clinical problem definition of prostate MRI.
Self-attention is an interesting strategy. Grouping it in different ISUP groupings is challenging.
Well described clinical problem definition of prostate MRI.
Self-attention is an interesting strategy. Grouping it in different ISUP groupings is challenging.
Well described clinical problem definition of prostate MRI.
Self-attention is an interesting strategy. Grouping it in different ISUP groupings is challenging.
Well described clinical problem definition of prostate MRI.
Self-attention is an interesting strategy. Grouping it in different ISUP groupings is challenging.


**Justification Of Rating:**

Well written paper

Well described clinical problem definition of prostate MRI.
Self-attention is an interesting strategy. Grouping it in different ISUP groupings is challenging.
Well described clinical problem definition of prostate MRI.
Self-attention is an interesting strategy. Grouping it in different ISUP groupings is challenging.
Well described clinical problem definition of prostate MRI.
Self-attention is an interesting strategy. Grouping it in different ISUP groupings is challenging.
Well described clinical problem definition of prostate MRI.
Self-attention is an interesting strategy. Grouping it in different ISUP groupings is challenging.


**Paper Type:**

methodological development

**Questions To Address In The Rebuttal:**

Miss statistical analysis


Well described clinical problem definition of prostate MRI.
Self-attention is an interesting strategy. Grouping it in different ISUP groupings is challenging.
Well described clinical problem definition of prostate MRI.
Self-attention is an interesting strategy. Grouping it in different ISUP groupings is challenging.
Well described clinical problem definition of prostate MRI.
Self-attention is an interesting strategy. Grouping it in different ISUP groupings is challenging.
Well described clinical problem definition of prostate MRI.
Self-attention is an interesting strategy. Grouping it in different ISUP groupings is challenging.


**Special Issue:**

no

---

### Official Review · AnonReviewer3 · 2020-03-17
**Complete description of method and experiments**

**Rating:** 3
**Confidence:** 4
**Recommendation:** Poster

**Summary:**

key ideas:
- multi-class deep network for to first segmentation of  PZ , second detect PZ prostate lesions and third GGG grading
- Input is bi-parametric MRI (ADC and T2w) in two separate decoding branches
experiments, and their significance:
Performance was evaluated using a large multivendor dataset correlating with  hysto-pathology as ground truth,  training and testing with a 5-fold cross-validation.
Adequate FROC analysis and kappa statistics was conducted to evaluate the performance

**Strengths:**

The dataset consists of a multi-vendor bi-parametric MRI collection acquired from prostate cancer patients.
FROC analysis and kappa statistics were performed to evaluate the performance
Method is clearly written


**Weaknesses:**

Only PZ cancer, why not includes TZ?
Please provide the standard deviation of the kappa statistics.
How does this performance compares the PROSTATEx challenge: https://prostatex.grand-challenge.org/

**Detailed Comments:**

1) Please include the standard deviation of the kappa statistics to understand the variation in your dataset.
2) The dataset consists of a multi-vendor bi-parametric MRI collection acquired from prostate cancer patients, is this from a single institute?

**Justification Of Rating:**

It is a good paper, but I miss the application towards the whole prostate, not just the PZ.
I miss the performance evaluation of the method on a public dataset such that the performance is comparable to existing methods in literature

**Paper Type:**

both

**Questions To Address In The Rebuttal:**

1) Please change to MRI wording into bi-parametric MRI as only ADC maps and T2-w images were used as input
2) In your method it is described that batch normalization is performed. Did you experiment the impact on the different MR machines? Did you consider other types of normalization?
3) Why was only PZ lesions included? 30% of prostate cancers are in the transition and anterior zones and are often missed or miss-classified by the radiologist.
4) I miss comparison to latest research especially from MIDL. How does your algorithm reflect to e.g "Simultaneous Detection and Grading of Prostate Cancer in Multi-Parametric MRI" from Vente et al. MIDL 2019 (https://openreview.net/forum?id=r1lqYQvAY4)



**Special Issue:**

no

---

> ### Author Response · Authors · 2020-03-27
> **Reply to the reviewer - Rebuttal**
>
> 1) Please provide the standard deviation of the kappa statistics
>
> Good point.  Here they are:
> kappa_panet_prostate = 0.347 土 0.052
> kappa_unet_prostate = 0.313 土 0.078
>
> 2) The dataset from a single institute?
>
> Yes.
>
> 3) Please change to MRI wording into bi-parametric MRI
>
> The modified version of the manuscript will include “bi-parametric MRI”.
>
> 4) Impact of batch normalization on the different MR machines. Did you consider other types of normalization?
>
> Besides batch norm, we pre-process the input images with a z-score normalization (zero mean and unit standard deviation). During training, batch normalization is applied as stated in section 2.2 (and as in Vente et al.). We did not notice any difference in performance among the two types of data in the validation fold, so we did not consider more elaborated transfer learning strategies.
>
> 5) Why was only PZ lesions included?
>
> Good question, we focused on PZ lesions as they account for 70% of prostate cancers.  Furthermore, the diagnosis of cancer in the TZ relies on different criteria than those in the PZ and this slightly goes beyond the scope of our dataset.  Our PZ approach is in par with several papers in the litterature (e.g. “Multiparametric MRI and auto-fixed volume of interest-based radiomics signature for clinically significant peripheral zone prostate cancer” by Bleker et al.).
>
> 6) How does your algorithm reflect to paper by Vente et al. 2019
>
> Vente et al. performs multi-class segmentation of PCa cancer with a UNet model similar to the one we used to compare our method.  Some methodological differences remain but we might be able to compare to their study. Supposing that the voxel-wise quadratic-weighted κ score defined in their Table 1 is computed considering the 5 GGG (including the GGG 1 where background, healthy prostate and GS 6 are gathered), we recalculated our kappa score. With our U-Net (which is more or less the same network as de Vente et al.’s), we got κ_unet = 0.463 土 0.033 and with ProstAttention-Net got k_panet = 0.538 土 0.032. This seems better than the voxel-wise quadratic-weighted κ score obtained by their SLOR method (0.391 ± 0.062).
>
> For a better comparison, we should test our architecture on the PROSTATEx-2 dataset using the same GG categorization. This point has been added to the discussion of the revised manuscript.

---

> > ### Comment · AnonReviewer3 · 2020-04-05
> > **Reply to rebuttal**
> >
> > I am happy with the clear answers. Thank you. Please adjust the rating as 4

---

### Official Review · AnonReviewer2 · 2020-03-19
**Multi-class classication of prostate cancer**

**Rating:** 1
**Confidence:** 5

**Summary:**

This paper proposes a U-Net based architecture which segments the prostate peripheral zone (PZ) and performs detection and multi-class classification of  PZ lesions.  The network includes an attention mechanism that allows searching only in PZ areas. The method was tested on a dataset 98 patients all included with T2w images, and ADC maps and the results were compared to those obtained by standard U-Net architecture trained without the attention mechanism. FROC analysis was used to evaluate the proposed method. The authors reported a 75.8% sensitivity at 2.5 false positives per patient and 66% sensitivity at 2.5 false positives per patient for their method and U-Net baseline model, respectively.

**Strengths:**

- The proposed architecture obtains the PZ segmentation together with lesion detection and lesion grading. Using the information on the lesion location improves the results by decreasing the number of false positives and by taking into account lesion differences among different areas


**Weaknesses:**

- A very similar approach has already been presented at MIDL 2019 (Effect of Adding Probabilistic Zonal Prior in Deep
Learning-based Prostate Cancer Detection) and not referenced
- The experimental part is not robust enough.
- The comparison with the baseline U-NEt is not fair; most of the CAD systems include the step of prostate segmentation. If the authors wanted to show how zonal segmentation improves CAD performance, they should have compared the results with U-Net trained on the whole prostate and not on the whole image.
- The lesion grade analysis is "to be considered with care" (as stated by the authors) due to the small number of lesions per class and fold.

**Justification Of Rating:**

The method does not have enough novelty, the authors did not reference a very similar paper. The statement of lesion grading is not supported by enough data and the experimental part lack of robustness. The authors did not show enough results to support their claim.

**Paper Type:**

validation/application paper

**Questions To Address In The Rebuttal:**

- What is the novelty proposed? What are the differences with the paper Effect of Adding Probabilistic Zonal Prior in Deep
Learning-based Prostate Cancer Detection?
-Why is the FROC on GS group for U-net not shown?
-The authors could show the comparison with FROC-AUC for both tasks
-Why did the authors not add a ROC analysis as well?
-"ProstAttention-Net performs similarly to FocalNet" this sentence should be supported by numbers. Authors should mention the values of sensitivity/specificity and FROC-AUC achieved by FocalNet
-Why Unet-baseline also finds PZ areas in Fig. 3?
-How is the grading taken into account in the loss?
- How the authors tackled the problem of ADC-T2w misalignment?

**Special Issue:**

no

---

> ### Author Response · Authors · 2020-03-27
> **Reply to the reviewer - Rebuttal**
>
> 1) What are the differences with the paper by Hosseinzadeh et al.? What is the novelty proposed?
>
> Sorry for missing that paper.  Like us, Hosseinzadeh et al. use the peripheral zone (PZ) to orient the focus of a tumor segmentation CNN. However, their methodology is different as they used two cascaded CNNs : one to segment the PZ and transition zones (TZ) and a second to segment PI-RADS 4 and 5 PCas lesions (binary segmentation). These two CNNs are trained separately. Our ProstAttention-Net is an end-to-end network (c.f Fig 2) which jointly performs the tasks of PZ segmentation (upper branch) and PZ + PZ lesion segmentation (lower branch). This network is trained with a combined loss reported in Eq.(1). Also, Hosseinzadeh et al.’s system is a binary classification of clinically significant PCas lesions while we perform a 7-class segmentation based on the Gleason score, i.e background, PZ, GS 6, GS 3+4, GS 4+3, GS8 and GS >=9. The characterization of PCa lesions aggressiveness constitutes one of our work’s main novelty.
>
> Immediate comparison with Hosseinzadeh et al.’s is not possible without reimplementing their method since we do not use the same lesion definition. Hosseinzadeh et al. consider PI-RADS 4 and PI-RADS 5 while we consider lesions with Gleason score (GS) > 6 as CS lesions. Also, PI-RADS score is defined by the radiologists and might not be representative of CS lesions.
>
> Let us also emphasize that we use whole-mount histopathology after prostatectomy groundtruths, the most accurate grading method to our knowledge. With that method, all lesions are included, even prospective lesions missed by the radiologists.
>
> Hosseinzadeh et al.’s paper will be cited and discussed in the revised version of our manuscript
>
>  2) Why is the FROC on GS group for U-net not shown? Authors could show the comparison with FROC-AUC for both tasks. Why not a ROC analysis as well?
>
> We put the class-wise FROC curve to underline the performances of our system wrt tumor grades and thought that Fig. 4(a) was enough to compare our method  to UNet.  That said, we do have class-wise UNet FROC curves and  will add it to the revised manuscript.  In short, at 1FP per patient, the sensitivity for the UNet vs our method is :
> GS>=9,             GS=8,           GS=4+3,          GS=3+4           GS=3+3
> 0.7 vs 0.8     0.43 vs 0.28    0.4 vs 0.48      0.43 vs 0.46,    0.17 vs 0.19
>
> Thus, except for GS=8, our method outperforms the UNet.
> We decided to use FROC curves and not ROC curves as it is more adapted to region based analysis such as our’s.
>
>  3) "ProstAttention-Net performs similarly to FocalNet" this sentence should be supported by numbers. Authors should mention the values achieved by FocalNet.
>
> For a 1FP per patient, we get ~40% sensitivity (c.f.Fig 4(a)) while Cao et al. report ~60% for a crossentropy loss and ~80% for a focal loss.  However, for a fair comparison, we need to update our evaluation protocol to fit their’s :  Cao et al. evaluate their method on slices with at least one lesion.  If we recalculate our results with that same slice selection criterions, we reach a sensitivity of ~65% at 1 FP and ~77% for a 1.5 FP which is quite good considering that our dataset contains 4 times fewer subjects.
>
> 4) Why Unet-baseline also finds PZ areas in Fig. 3?
>
> We wanted to have a comparison as fair as possible between our method and UNet.  Like our method, UNet was trained to segment 7 classes, i.e. background, PZ, GS 6, GS 3+4, GS 4+3, GS8 and GS >=9 (answer to weakness #3). In this way, results clearly show that the improvement of our approach comes from its attention mechanism.
> Let us also mention that for UNet and our method, the input image and the output maps were not masked with a prostate segmentation. Thus, both methods were trained on the whole image (answer to weakness # 3).
> The manuscript will be reworded to clarify these points.
>
>  5) How is the grading taken into account in the loss?
>
> Our ProstAttention-Net is a 7 classes segmentation model, where each grading corresponds to a class (our grading classes have no ordinal ordering). The overall loss includes two losses: 1) the cross entropy between predictions and ground truth 2) the generalised dice loss, as a class-weighted dice loss.
>
> 6) How the authors tackled the problem of ADC-T2w misalignment?
>
> Images in our dataset are fairly well aligned.  We tested our system on raw input images as well as on registered images and found no significant differences (Cao et al. came to the same conclusion).

---

> > ### Comment · AnonReviewer2 · 2020-04-06
> > **Rebuttal**
> >
> > - The sentence "While we perform a 7-class segmentation based on the Gleason score, i.e. background, PZ, GS 6, GS 3+4, GS 4+3, GS8 and GS >=9. The characterization of PCa lesions aggressiveness constitutes one of our work’s main novelty."  is in contrast with this statement  "The lesion grade analysis is to be considered with care (as stated by the authors) due to the small number of lesions per class and fold." It seems that the results of the proposed main novelty are not to be considered.
> > -  No comparison with FROC-AUC is given. I do not think one-point comparison can provide enough information also for the small differences that the results of both methods achieved.
> >
> > I still think the paper does not show enough novelty and more importantly, convincing results.

---

> > > ### Author Response · Authors · 2020-04-06
> > > **Rebuttal**
> > >
> > > 1) The network is intrinsically multiclass and thus outputs a gleason score grade. There is no binary classification at any time. The consideration of lesion aggressiveness is therefore included in all of our results : kappa score, FROC for CS cancer detection, FROC by GS group.
> > > Following the reviewer suggestion (please see our response to comment #2), we decided to add class-wise FROC curves for the UNet and our method to the revised manuscript. In short, at 1FP per patient, the sensitivity for the U-Net vs our method is
> > > GS>=9,             GS=8,           GS=4+3,          GS=3+4           GS=3+3
> > > 0.7 vs 0.8     0.43 vs 0.28    0.4 vs 0.48      0.43 vs 0.46,    0.17 vs 0.19
> > >
> > > Thus, except for GS=8, our method outperforms the UNet. This confirms the novelty and performance of our method.
> > >
> > > 2) FROC is different from ROC curve as it is unbounded. This is why FROC-AUC can not be computed as it would be meaningless.
> > >
> > > We reach 75.8% of sensitivity at 2.5 false positive per patients on a 24-slices volume in a 7-classes segmentation task.  If we consider only slices with at least one ground truth lesion, we reach ~77% for a 1.5 FP. Those results are competitive with regards to state-of-the-art performance.
> > > We are not performing binary segmentation (CS cancer versus non cancer) unlike most of the papers performing PCa segmentation.

---

### Author Response · Authors · 2020-04-02
**Comments and further questions regarding the rebuttal**

Dear reviewers,

we would like to know if you had time to look at our answers to your comments. We would be happy to know what you think about it, if you have any further questions that we could answer and if our answers had an influence on your rating.

yours truly,

The authors

---

### Meta-Review · Area_Chair1 · 2020-04-05
**MetaReview of Paper238 by AreaChair1**

**Rating:** 3

**Metareview:**

The paper appears to not convincingly add much to existing work or related work that was mostly overlooked. Overall well written, the paper may have limited clinical applicability due to the restriction to specific cancer types.  However, discussion has shown that the paper could lead to interesting discussions and further fruitful exchanges if presented at the conference. Clarification and inclusion of the points made by the reviewers are required for acceptance.

**Paper Type:**

methodological development

**Special Issue:**

no

---

> ### Author Response · Authors · 2020-04-06
> **MetaReview of Paper238 by AreaChair1**
>
> Dear Area Chair, thank you for your encouraging recommandation. We will undertake to clarify and include the points made by the reviewers in the accepted version of our paper.
> The authors

---

### Decision · Program_Chairs · 2020-04-11

Accept